Insights into soil nematode diversity and bacterial community of Thai jasmine rice rhizosphere from different paddy fields in Thailand

Nimnoi Pongrawee 1
Pirankham Patawee 2
Srimuang Kittipong 3
Ruanpanun Pornthip pornthip.r@ku.th 2
1 Microbiology Division, Department of Science and Bioinnovation, Faculty of Liberal Arts and Science, Kasetsart University, Kamphaeng Saen Campus , Kamphaeng Saen , Nakhon Pathom , Thailand
2 Department of Plant Pathology, Faculty of Agriculture at Kamphaeng Saen, Kasetsart University, Kamphaeng Saen Campus , Kamphaeng Saen , Nakhon Pathom , Thailand
3 Prachinburi Rice Research Center, Division of Rice Research and Development, Rice Department , Ban Sang , Prachin Buri , Thailand
Ahmad Faheem
Electronic publication date: 2024 Apr 23
Publication date: 2024
Volume: 12
Electronic Location ID: e17289
Received 2023 Dec 21; Accepted 2024 Apr 2
Copyright: ©2024 Nimnoi et al.
Copyright year: 2024
Copyright holder: Nimnoi et al.
License: This is an open access article distributed under the terms of the Creative Commons Attribution License, which permits unrestricted use, distribution, reproduction and adaptation in any medium and for any purpose provided that it is properly attributed. For attribution, the original author(s), title, publication source (PeerJ) and either DOI or URL of the article must be cited.
License URL: https://creativecommons.org/licenses/by/4.0/

Keywords: Plant-parasitic nematode, Oryza sativa L., Metagenome, Bacteriome

Funding: National Research Council of Thailand (NRCT) This project is funded by the National Research Council of Thailand (NRCT). The funders had no role in study design, data collection and analysis, decision to publish, or preparation of the manuscript.

==============================
Globally, phytonematodes cause significant crop losses. Understanding the functions played by the plant rhizosphere soil microbiome during phytonematodes infection is crucial. This study examined the distribution of phytonematodes in the paddy fields of five provinces in Thailand, as well as determining the keystone microbial taxa in response to environmental factors that could be considered in the development of efficient biocontrol tactics in agriculture. The results demonstrated that Meloidogyne graminicola and Hirschmanniella spp. were the major and dominant phytonematodes distributed across the paddy fields of Thailand. Soil parameters (total P, Cu, Mg, and Zn) were the important factors affecting the abundance of both nematodes. Illumina next-generation sequencing demonstrated that the levels of bacterial diversity among all locations were not significantly different. The Acidobacteriota, Proteobacteria, Firmicutes, Actinobacteriota, Myxococcota, Chloroflexi, Verrucomicrobiota, Bacteroidota, Gemmatimonadota, and Desulfobacterota were the most abundant bacterial phyla observed at all sites. The number of classes of the Acidobacteriae, Clostridia, Bacilli, and Bacteroidia influenced the proportions of Hirschmanniella spp., Tylenchorhynchus spp., and free-living nematodes in the sampling dirt, whereas the number of classes of the Polyangia and Actinobacteria affected the amounts of Pratylenchus spp. in both roots and soils. Soil organic matter, N, and Mn were the main factors that influenced the structure of the bacterial community. Correlations among rhizosphere microbiota, soil nematodes, and soil properties will be informative data in considering phytonematode management in a rice production system.

Introduction

Presently, a major challenge for agro-industrial operations is identifying the way to secure food for the world’s citizens, which at its current rate of increase is estimated to reach nearly 10 billion. While global food demand is expected to double by 2050, as the availability and quality of natural resources that support agricultural practices continue to diminish, their scarcity and degradation will become more pronounced (United Nations, 2022). Rice, as a staple nourishment, holds significant importance as a global food for humankind and is extensively traded and captivated in over 110 countries globally (Corrêa et al., 2007; Gnamkoulamba et al., 2018; Musarrat et al., 2016). In recent years, the virulence of phytonematodes has increased due to the changing climate. This has caused worsening losses in rice production (Mantelin, Bellafiore & Kyndt, 2017). Parasitic nematodes of rice can adapt well and survive in flooded and irrigated conditions (Fernandez, Cabasan & De Waele, 2014; Win et al., 2013). Among the nematodes known as pathogens of rice, only 29 species have caused considerable yield losses (Bridge, Plowright & Peng, 2005). Criconema spp., Dolichodorus spp., Helicotylenchus spp., Heterodera spp., Hirschmanniella spp., Meloidogyne spp., Pratylenchus spp., Scutellonema spp., Tylenchorhynchus spp., and Xiphinema spp. have been predominantly found in the rice rhizosphere and roots (Gnamkoulamba et al., 2018; Musarrat et al., 2016; Shahabi et al., 2016). Damage levels in both direct and indirect ways in rice caused by phytonematodes result in deferred plant growth and maturation, stunting, yellowing, and grain yield loss that subsequently reduce the income (Onkendi et al., 2014; Sharma Poudyal et al., 2005; Win, Kyi & De Waele, 2011). Phytonematode infestations around the world create an estimated annual yield loss up to USD 150 billion (Singh, Singh & Singh, 2015).

To cope with this problem, the relationship among phytonematodes, the microbial community, and physiochemical soil characteristics is one promising and important way to provide useful information for the implementation of an appropriate soil management program to limit harmful nematodes to crops. Previous research demonstrated that many physiochemical soil traits, such as pH, soil texture, soil elements, and organic matter (OM), are correlated with the existence and density of nematodes, as well as with bacterial diversity and bacterial community levels (Huang et al., 2020; Renčo, Gömöryová & Čerevková, 2020; Zhou et al., 2019). Bacterial communities have been revealed to suppress nematode infection with mechanisms that regulate nematode population densities (Zhou et al., 2019). Although the impact of soil-borne nematodes on plant-associated microbial communities has been theorized to play significant roles in plant development and yield (Markalanda et al., 2022; Pieterse, De Jonge & Berendsen, 2016; Zhou et al., 2019), less is understood about the specific relationships between phytonematodes and the microbiome in root and of rice. Recently, in Vietnam, Masson et al. (2020) studied the microbiome of infected and uninfected rice roots for Meloidogyne graminicola (rice root-knot nematode). They revealed that M. graminicola infection results in the huge restructuring of bacterial communities by influencing species richness and taxa abundance.

To our knowledge, no research has looked at the association between the phytonematode population density and the rice root-associated bacteriome from the same rice cultivar under different natural environmental conditions, especially in Thailand. Hence, this study’s goals were to investigate the bacterial diversities and communities of the rhizosphere soil from Thai Hom Mali rice (Oryza sativa L.) cv. Khao Dawk Mali 105 (KDML105) from five natural paddy fields in Thailand and to elucidate the relationships between phytonematode populations, root-associated bacterial communities, and soil physicochemical factors. The massive data generation and the derived comprehensive versatile knowledge may aid soil management as well as resource conservation in agroecosystems of economic importance.

Materials & Methods

Soil sample collection

During September 2022, surveillance was conducted of the phytonematode distribution in five notable paddy fields of KDML105 in tillering stage (45 to 55-day plant) in Chachoengsao, CCS (13°36′46.1″N 101°16′54.3″E); Nakhon Nayok, NYK (14°16′25.4″N101°08′37.0″E); Prachin Buri, PAR (14°10′01.3″N101°35′23.1″E); Pathum Thani, PTT (14°09′32.7″N 100°43′55.4″E); and Phra Nakhon Si Ayutthaya, AYY (14°26′43.6″N 100°45′52.5″E). In each field, three sampling areas were fixed randomly to collect soil samples, depending on the field size. The sampling area measuring 20 × 20 m was methodically subdivided into a grid pattern of 5 × 5 m mesh lines to facilitate sample collection. In total, 16 subsamples of rice roots and their rhizosphere soils depth of 15 cm were collected from 16 points of the grid area, and then the soils were meticulously combined to create a composite sample weighing 10 kg, which aimed to represent the overall soil composition throughout the study spot (Win, Kyi & De Waele, 2011). The samples were carefully enclosed in individual plastic pouches, securely sealed, and promptly delivered to the Agricultural Nematology and Microbiology Laboratory for investigation. The soil samples were partitioned into three parts: 3 kg for nematode extraction, 5 kg were air-dried for physical and chemical property analysis, and the last 2 kg were kept at 4 °C for assessing the bacteriome.

Isolation and identification of nematodes

The bulk was extracted from rice root following the method described by Barillot et al. (2013). A modified Baermann tray technique was used for extracting nematodes from the bulk (Schindler, 1961). The rice roots were washed carefully under running water to eliminate all traces of dirt particles, meticulously chopped into small pieces (1–2 cm), and then blended in 0.8% NaOCl for 30 s. For 10 min, the mixes were left at room temperature before applying a modified Baermann funnel method (Hooper, 1986). Nematode suspensions derived from soil and root samples were obtained following a 48-h incubation period and subsequently subjected to inspection using a stereomicroscope. (Olympus SZ51; Tokyo, Japan). The nematodes without any stylets were identified as free-living nematodes. The identification of phytonematodes was conducted at the genus level through the analysis of morphological traits (Tarjan, Esser & Chang, 1977). The body size, stylet length, and vulva position were measured using the CellSens imaging software (V1.6) with an Olympus DP26 camera (Tokyo, Japan). Meloiodogyne species were identified using the perineal pattern of adult females (Hunt & Handoo, 2009) and species-specific primers (Mg-F3 5′-TTATCGCATCATTTTATTTG-3′ and Mg-R2 5′-CGCTTTGTTAGAAAATGACCCT-3′) as described by Htay et al. (2016).

Soil physical and chemical property analysis

Dried soil samples were passed through a sieve (2 mm and 10 mesh size). The pH and electrical conductivity (Ec) were examined (Slavich & Petterson, 1993). Using a pipette-based technique, the sample’s particle-size distribution was evaluated (Gee & Bauder, 1986). The amount of OM in the soil was measured (Walkley & Black, 1934). The analysis of the available N was conducted (Guebel, Nudel & Giulietti, 1991). The quantity of total P was analyzed (Bray & Kurtz, 1945). The amounts of Ca, K, and Mg were estimated (Del Castilho & Rix, 1993). Elements including, Cu, Fe, Mn, and Zn were also quantified (Lindsay & Norvell, 1978).

DNA isolation and next generation sequencing

The rhizosphere soils were extracted using the standardized protocol as described by Barillot et al. (2013). Total DNA extraction was performed on three duplicates of rhizosphere soils from each location by a DNA soil extraction kit of NucleoSpin (Macherey-Nagel, Germany). The amplification of 16S rDNA was performed as follows Apprill et al. (2015) and Parada, Needham & Fuhrman (2016). The PCR results were cleaned up by Vivantis gel extraction kit (Vivantis, Malaysia). The amplified DNA libraries were created and determined using Illumina-HiSeq2500 (Illumina, San Diego, CA, USA). Through parallel amplification and sequencing, negative controls (reactions with sterile water) were performed.

Bioinformatics analyses

The FLASH program was employed to combine raw reads (Magoč & Salzberg, 2011). The raw reads were screened by the QIIME program for high-quality sequences (Caporaso et al., 2010). The UCHIME program was employed for determination and elimination of chimera (Haas et al., 2011). The Uparse program was applied to perform the clustering and species annotation of operational taxonomic unit (OTU) (Edgar, 2013). The Mothur program was performed to annotate bacterial species (Quast et al., 2013; Schloss et al., 2009). All OTUs obtained from representative reads were analyzed phylogenetically using the MUSCLE tool (Edgar, 2004).

Statistical determination

The beta and alpha diversity parameters as well as sequencing depth were computed with the QIIME program (Caporaso et al., 2010). The R program was used to display the analyzed data (R Core Team, 2013). For determination and reducing dimensionalities of data, the PCoA and NMDS were used. The QIIME program was used to determine the similarity between samples by the UPGMA method (Caporaso et al., 2010). The LEfSe analysis was computed by the LEfSe program to discover high-dimensional biomarkers between samples (Segata et al., 2011). The AMOVA and ANOSIM were computed to disclose the significant differences between bacterial communities. The PAST program was employed for canonical correlation analysis (CCA) (Hammer, Harper & Ryan, 2001). The ANOVA with Tukey’s test was employed to signify the alpha diversity indices, phytonematodes, free-living nematodes, and soil physicochemical parameters. Based on Spearman’s correlation, the relationships between soil physicochemical factors and phytonematodes, free-living nematodes, and bacterial populations were clarified. The Spearman’s between-group analysis as well as ANOVA were computed by the SPSS program (IBM, Armonk, NY, USA).

Results

Diversity of soil nematodes associated with Oryza sativa L.

The diversity of phytonematodes and free-living nematodes was determined within the rhizosphere soils and roots of the rice collected from five paddy fields of notable provinces in Thailand (Table S1). Based on the morphological characteristics of extracted nematodes, free-living nematodes, and phytonematodes comprised of Meloidogyne, Hirschmanniella, Pratylenchus, Helicotylenchus, Tylenchorhynchus were found (Fig. 1). Identification of Meloidogyne species, we found the perineal pattern of adult females was rounded, with smooth striae and no lateral field. These perineal features were like the pattern described by M. graminicola. Molecular method confirmation by using species-specific primer, the expected size (∼369 bp) of the PCR product for M. graminicola was detected (Fig. S1). Among all the rhizosphere soils, site PTT had the highest number of M. graminicola (141.66 ± 38.10 nematodes/500 g soil), which was significantly different compared to the other sites, followed by sites CCS, PAR, AYY, and NYK, respectively (Table 1). Site PTT also had the highest number of Hirschmanniella spp. (22.00 ± 4.58 nematodes/500 g soil), which was significantly different compared to the other sites. Pratylenchus spp. were found only at sites PAR (11.00 ± 3.60 nematodes/500 g soil) and NYK (4.66 ± 0.57 nematodes/500 g soil). Helicotylenchus spp. were found only at sites PTT and CCS, with site PTT having a significantly higher number compared to site CCS. Sites CCS and NYK were the only ones where Tylenchorhynchus spp. were identified; however, their numbers did not differ from each other. The number of free-living nematodes was highest at site AYY (117.66 ± 29.36 nematodes/500 g soil), which was a significant difference compared to the other sites.

Figure 1 Identification of nematodes obtained from roots and rhizosphere soils of Oryza sativa L. cv. Khao Dawk Mali 105 based on morphology.

Second-stage juveniles of M. graminicola (A), adult females of Hirschmanniella (B), Pratylenchus (C), Helicotylenchus (D), Tylenchorhynchus (E), free-living nematodes (F–G) and perineal pattern of M. graminicola (I).

Besides the phytonematodes and free-living nematodes in the rhizosphere soils, we also examined their presence within roots. The findings demonstrated that site CCS had the most M. graminicola (3,280.00 ± 603.98 nematodes/3 g root), which was noticeably different from the other sites, followed by sites AYY, PTT, PAR, and NYK, in that order, respectively (Table 2). The number of Hirschmanniella spp. was highest at site AYY and was not significantly different compared to the other sites. Pratylenchus spp. were found at sites PAR (9.33 ± 2.51 nematodes/3 g root) and NYK (6.33 ± 3.05 nematodes/3 g root). Helicotylenchus spp. and Tylenchorhynchus spp. were detected at the PTT and CCS sites, respectively. The PAR site had the greatest density of free-living nematodes. This study demonstrated that the rhizosphere of rice at PTT province was confronted with severe epiphytotic levels of M. graminicola, Hirschmanniella spp., and Helicotylenchus spp., whereas the greatest concern in regarding the distribution of free-living nematodes in the rhizosphere soil was at AYY.

Table 1 Soil nematodes from rhizosphere collected from rice fields.

Sampling site	No. of plant-parasitic nematode/500 g soil*	No. of free-living nematode/ 500 g soil*	
	M. graminicola	Hirschmanniella spp.	Pratylenchus spp.	Helicotylenchus spp.	Tylenchorhynchus spp.		
CCS	39.33 ± 15.04a**	7.33 ± 7.50a	0.00	9.66 ± 9.86a	5.00 ± 1.73a	39.66 ± 34.15a	
NYK	18.00 ± 5.00a	4.33 ± 2.30a	4.66 ± 0.57a	0.00	4.66 ± 2.08a	26.33 ± 3.05a	
PTT	141.66 ± 38.10b	22.00 ± 4.58b	0.00	46.33 ± 16.25b	0.00	51.33 ± 20.55a	
AYY	34.00 ± 9.84a	9.00 ± 3.60a	0.00	0.00	0.00	117.66 ± 29.36b	
PAR	36.00 ± 10.00a	9.33 ± 4.16ab	11.00 ± 3.60b	0.00	0.00	48.00 ± 9.00a	
Notes.

* All values are represented as (mean ±SD) based on triplicate samples.

** Values with the same letters in the column are not significantly different (P>0.05) according to Tukey’s test.

CCS sampling from Chachoengsao province

NYK sampling from Nakhon Nayok province

PTT sampling from Pathum Thani province

AYY sampling from Phra Nakhon Si Ayutthaya province

PAR sampling from Prachin Buri province

Table 2 Soil nematodes in roots collected from rice fields.

Sampling site	No. of plant-parasitic nematode/3g root*	No. of free- living nematode /3 g root*	
	M. graminicola	Hirschmanniella spp.	Pratylenchus spp.	Helicotylenchus spp.	Tylenchorhynchus spp.		
CCS	3,280.00 ± 603.98c**	6.00 ± 4.35a	0.00	0.00	10.00 ± 8.18a	0.00	
NYK	38.66 ± 8.14a	13.00 ± 4.58a	6.33 ± 3.05a	0.00	0.00	3.33 ± 1.52a	
PTT	307.66 ± 76.10a	10.00 ± 8.18a	0.00	6.66 ± 3.78a	0.00	4.00 ± 2.00a	
AYY	1,846.00 ± 406.73b	14.33 ± 3.51a	0.00	0.00	0.00	0.00	
PAR	116.00 ± 24.43a	11.66 ± 3.51a	9.33 ± 2.51a	0.00	0.00	16.66 ± 5.68b	
Notes.

* All values are represented as (mean ± SD) based on triplicate samples.

** Values with the same letters in the column are not significantly different (P > 0.05) according to Tukey’s test.

CCS sampling from Chachoengsao province

NYK sampling from Nakhon Nayok province

PTT sampling from Pathum Thani province

AYY sampling from Phra Nakhon Si Ayutthaya province

PAR sampling from Prachin Buri province

Soil parameters

The soil physicochemical parameters of each sampling site were characterized. All observed locations, pH, and Ec values were in the ranges of 4.77–7.81 and 0.37–10.52 ds/m, respectively (Table 3). Site AYY had the highest soil pH, whereas site PAR had the lowest soil pH. The Ec value for site CCS was the highest, while the lowest was at site NYK. Site AYY had the highest amounts of OM (4.41 ± 0.35%) and available N (0.21 ± 0.01%), which were significantly different compared to the other sites, followed by sites PTT, NYK, PAR, and CCS, respectively. The amounts of total P at each sampling site differed significantly from each other. Site CCS had the highest concentration of total K (235.93 ± 2.37 mg/kg), with both being significantly different compared to the other sites. Total Ca (5301.19 ± 99.21 mg/kg) was significantly the highest at site AYY, whereas it was significantly the lowest at site PAR (381.06 ± 27.27 mg/kg). Site PTT had the highest amount of total Mg (674.91 ± 3.19 mg/kg), which was significantly different compared to the other sites. Total Fe (379.33 ± 4.86 mg/kg) was significantly the highest at site PAR. The amounts of total Mn at each sampling site differed significantly from each other. Site AYY had the greatest amount of total Mn (61.78 ± 0.32 mg/kg), followed by sites PTT, PAR, NYK, and CCS. Site PTT had the highest amounts of total Cu (46.12 ± 0.70 mg/kg) and Zn (15.51 ± 0.86 mg/kg), which differed significantly from the other sites. The pH value and amounts of OM, available N, total Ca, and Mn at site AYY were the greatest. While concentrations of total P, Mg, Cu, and Zn at site PTT were highest. Site CCS presented the highest Ec value and amount of total K. The amount of total Fe was significantly highest at site PAR.

Table 3 Soil physicochemical properties from each sampling site.

Parameter	Sampling site	
	CCS	NYK	PTT	AYY	PAR	
pH	5.94	5.21	6.53	7.18	4.77	
Electrical conductivity (ds m−1)	10.52	0.37	0.84	0.79	0.42	
Sand (%)	59.20	47.17	6.00	11.10	35.61	
Silt (%)	32.19	40.49	34.18	34.04	52.9	
Clay (%)	8.61	12.34	59.82	54.87	11.48	
Soil texture	Sandy Loam	Loam	Clay	Clay	Silt Loam	
Organic matter (%)*	0.51 ± 0.01a**	1.81 ± 0.26c	2.44 ± 0.49d	4.41 ± 0.35e	1.18 ± 0.25b	
Available N (%)*	0.02 ± 0.00a	0.09 ± 0.00c	0.11 ± 0.00d	0.21 ± 0.01e	0.07 ± 0.00b	
Total P (mg kg−1)*	18.57 ± 0.23b	8.49 ± 0.43a	48.48 ± 0.41e	26.23 ± 0.27c	29.64 ± 0.15d	
Total K (mg kg−1)*	235.93 ± 2.37e	11.98 ± 0.69b	228.05 ± 2.35d	185.51 ± 0.34c	15.23 ± 0.11a	
Total Ca (mg kg−1)*	3,044.04 ± 64.14c	518.55 ± 33.28b	3,138.17 ± 31.85c	5,301.19 ± 99.21d	381.06 ± 27.27a	
Total Mg (mg kg−1)*	633.41 ± 28.11d	7.11 ± 0.12a	674.91 ± 3.19e	397.76 ± 3.35c	29.38 ± 0.60b	
Total Fe (mg kg−1)*	116.76 ± 0.30a	230.37 ± 27.88c	177.65 ± 0.57b	238.90 ± 4.18c	379.33 ± 4.86d	
Total Mn (mg kg−1)*	2.29 ± 0.07a	8.25 ± 0.33b	59.99 ± 1.18d	61.78 ± 0.32e	16.08 ± 0.38c	
Total Cu (mg kg−1)*	1.69 ± 0.01b	0.45 ± 0.01a	46.12 ± 0.70d	7.26 ± 0.08c	0.93 ± 0.04ab	
Total Zn (mg kg−1)*	0.81 ± 0.02a	0.70 ± 0.00a	15.51 ± 0.86c	2.07 ± 0.05b	0.72 ± 0.02a	
Notes.

* All values are represented as (mean ±SD) based on triplicate samples.

** Values with the same letters in the column are not significantly different (P > 0.05) according to Tukey’s test.

CCS sampling from Chachoengsao province

NYK sampling from Nakhon Nayok province

PTT sampling from Pathum Thani province

AYY sampling from Phra Nakhon Si Ayutthaya province

PAR sampling from Prachin Buri province

Sequence analysis, bacterial diversity, and richness indices

The bacterial diversity and richness of all sampling sites were determined. In total, 1,955,800 raw sequences were acquired from fifteen DNA samples (three replicates/field). Tag merge and sequence quality control were performed to retrieve a total of 1,936,693 qualified tags (99.20% of the raw reads). In total, 1,263,601 taxon tags were obtained after removing the potential chimera tags. The tags with ≥97% resemblance were assigned to the same OTU. In total, 16,338 OTUs were obtained from all sampling sites with 98.79 ± 0.00% of Good’s coverage. Figure 2A displays the total tags, taxon tags, unclassified tags, unique tags, and OTU numbers for each replicate. A Venn diagram (Fig. 2B) was used to present the numbers of unique, common, and overlapping OTUs between sampling sites. This diagram showed 1,659 OTUs presented across all sampling sites. The greatest number of unique OTUs was found at site CCS (2,560), followed by sites PAR, AYY, PTT, and NYK, in that order. Additional analysis investigated the number of observed species, diversity as indicated by the Shannon-Weaver and Simpson indices, and richness as indicated using Chao1 and ACE for each sampling site (Table 4). The results demonstrated that site CCS presented the greatest number of observed species (5,260.00 ± 291.82), followed by sites PAR, PTT, NYK, and AYY, respectively. The higher values for the Shannon-Weaver indice implied greater bacterial diversity at site CCS, followed by sites PTT, AYY, PAR, and NYK, respectively, though the values did not differ significantly between all sites. Furthermore, the Chao1 and ACE values indicating bacterial richness illustrated that site CCS had the highest amount of bacterial richness, followed by sites PAR, PTT, and NYK, respectively, whereas site AYY had the lowest amount of bacterial richness. We found that site CCS had the highest numbers of unique OTUs and observed species, and the greatest diversity and richness of bacteria, whereas site NYK showed the lowest number of unique OTUs and the least bacterial diversity. The least number of detected bacterial species and richness were found at site AYY.

Figure 2 Tags and OTU numbers of each sampling location (A) and Venn diagram presenting the numbers of unique, common, and overlapping OTUs between each sampling site (B).

CCS, sampling from Chachoengsao province; NYK, sampling from Nakhon Nayok province; PTT, sampling from Pathum Thani province; AYY, sampling from Phra Nakhon Si Ayutthaya province; PAR, sampling from Prachin Buri province.

Table 4 Indices of bacterial richness and diversity of soil from each location.

Sampling site	Observed species*	Diversity indices*	Richness indices*	
		Shanon-Weaver	Simpson	Chao1	ACE	
CCS	5,260.00 ± 291.82b**	10.20 ± 0.32a	0.99 ± 0.00a	5,613.21 ± 414.84b	5,715.23 ± 408.28c	
NYK	4,023.66 ± 411.68ba	9.68 ± 0.24a	0.99 ± 0.00a	4,378.77 ± 484.18ab	4,424.89 ± 466.56ab	
PTT	4,205.66 ± 774.84ab	9.99 ± 0.54a	0.99 ± 0.00a	4,886.85 ± 569.25ab	5,024.60 ± 510.50abc	
AYY	3,644.00 ± 472.95a	9.87 ± 0.31a	0.99 ± 0.00a	3,963.25 ± 516.43a	4,094.70 ± 493.20a	
PAR	4,752.66 ± 260.34ab	9.82 ± 0.08a	0.99 ± 0.00a	5,318.71 ± 313.10b	5,438.49 ± 355.40bc	
Notes.

* All values are represented as (mean ± SD) based on triplicate samples.

** Values with the different letters in the column are significantly different (P ≤ 0.05) according to Tukey’s test.

CCS sampling from Chachoengsao province

NYK sampling from Nakhon Nayok province

PTT sampling from Pathum Thani province

AYY sampling from Phra Nakhon Si Ayutthaya province

PAR sampling from Prachin Buri province

NGS and bacterial communities

In all sampling sites, the phylum Acidobacteriota was most abundant (10.18–36.26%), followed by Proteobacteria (12.80–30.24%), Firmicutes (5.25–10.40%), Actinobacteriota (3.08–9.11%), Chloroflexi (4.27–9.01%), Myxococcota (2.12–8.84%), Verrucomicrobiota (3.36–7.33%), Bacteroidota (1.64–5.63%), Gemmatimonadota (1.43–5.33%), and Desulfobacterota (0.66–4.23%). Based on the biomarker analysis, the LDA score illustrated statistically unique communities at each sampling site. As depicted in Fig. S2A, there were differences in the bacterial community composition at each sampling site. The relative abundance of the phyla Chloroflexi, Gemmatimonadota, and Bacteroidota exhibited a statistically significantly increase at site PTT. The phyla Actinobacteriota and Cyanobacteria were more abundant at site PAR, whereas the Verrucomicrobiota were significantly abundant at site NYK, and similarly, the Nitrospirota at site CCS and both the Myxococcota and Desulfobacterota were the significantly predominant phyla at site AYY. However, there were also statistically distinguishable variations in the phyla among the samples. For example, the variability of the phylum Edwardsbacteria was shown to be statistically significant across the sites PTT, AYY, and PAR. Similarly, the phylum Latescibacterota exhibited considerable variability across the sites CCS, NYK, and PTT. The phylum Nitrospirota was variable among all sites significantly (Fig. S2B). Moreover, evaluating the top-10 predominant bacterial classes distributed in the soil at each sampling site (Table 5), site NYK had the highest numbers of Acidobacteriae and Verrucomicrobiae, which differed significantly from the other sites. The numbers of alpha-proteobacteria and Anaerolineae were highest at sites CCS and PTT, respectively. The diversity of Bacilli and Bacteroidia was highest at site PTT. Site AYY had the highest number of gamma-proteobacteria. Site PAR had the highest numbers of Clostridia and Actinobacteria. Sites AYY and PTT had the highest numbers of Polyangia, which differed significantly from other sites.

Table 5 Top 10 most abundance of bacterial classes presents in each observed location.

Sampling site	Class*	
	Acidobacteriae	gamma-proteobacteria	alpha-proteobacteria	Clostridia	Verrucomicrobiae	Bacilli	Bacteroidia	Anaerolineae	Polyangia	Actinobacteria	
CCS	17.35 ± 1.42c	18.68 ± 2.84b	8.11 ± 0.42c	3.25 ± 0.64a	2.57 ± 0.24a	1.49 ± 0.16a	3.39 ± 0.71b	2.39 ± 0.57a	1.43 ± 0.15a	1.49 ± 0.38a	
NYK	33.24 ± 4.00d	6.64 ± 0.39a	6.15 ± 0.03ab	4.71 ± 1.22a	7.19 ± 1.22c	3.35 ± 0.19a	1.47 ± 0.98a	1.31 ± 0.91a	1.41 ± 0.10a	2.96 ± 0.12ab	
PTT	5.82 ± 0.42a	15.99 ± 3.58ab	7.09 ± 1.11bc	5.41 ± 0.48a	4.74 ± 0.24b	4.74 ± 0.81a	5.30 ± 0.71c	5.64 ± 0.49b	4.53 ± 0.58b	2.82 ± 0.96ab	
AYY	4.76 ± 0.87a	24.30 ± 6.56b	4.50 ± 0.40a	5.66 ± 3.11a	4.83 ± 1.12b	4.63 ± 2.68a	4.64 ± 0.22bc	2.74 ± 0.94a	4.53 ± 0.85b	1.44 ± 0.73a	
PAR	11.85 ± 0.88b	19.03 ± 4.05b	11.19 ± 0.84d	5.70 ± 1.24a	3.80 ± 0.45ab	3.50 ± 0.51a	4.03 ± 0.50bc	1.47 ± 0.12a	0.95 ±.011a	4.14 ± 0.90b	
Notes.

* All values are represented as (mean ± SD) based on triplicate samples.

** Values with the different letters in the column are significantly different (P ≤ 0.05) according to Tukey’s test.

CCS sampling from Chachoengsao province

NYK sampling from Nakhon Nayok province

PTT sampling from Pathum Thani province

AYY sampling from Phra Nakhon Si Ayutthaya province

PAR sampling from Prachin Buri province

Heat map analysis was employed to determine more clearly the distribution at the genus level, which contributed to the structure of the community at each sampling site (Fig. 3). At site NYK, the more predominant genera were the Bryobacter, ADurb.Bin063-1 (Verrucomicrobia bacterium), Candidatus_Soilbacter, Candidatus_Koribacter, and Candidatus_Udaeobacter. The Bifidobacterium, WPS-2 (Eremiobacterota), WD2101 soil group (Planctomycetes), Acidibacter, Pseudomonas, Burkholderia, Caballeronia, Paraburkholderia, Pantoea, Faecalibacterium, and Bacteroides were the predominant genera at site PAR. The most abundant genera at site PTT were the Gemmatimoonas, Sphingomonas, and Latescibacterota. Thioalkalispira-Sulfurivermis, Sporacetigenium, Ellin6067 (beta-proteobacteria), Anaeromyxobacter, Mycoplasma, MND1, and Thiobacillus were the common genera at site AYY. The A21b (uncultured bacterium), Subgroup_13 (Acidobacteria), Subgroup_2 (Acidobacteria), RCP2-54 (uncultured bacterium), and Bacteriap25 (uncultured bacterium) were more abundant than other genus at site CCS.

Figure 3 Heat map analysis of distribution of genus in each sampling location.

CCS, sampling from Chachoengsao province; NYK, sampling from Nakhon Nayok province; PTT, sampling from Pathum Thani province; AYY, sampling from Phra Nakhon Si Ayutthaya province; PAR, sampling from Prachin Buri province.

The AMOVA results showed major variations in the structure of the community among the sites (Fs = 14.22; P < 0.001). In addition, the inter- and inner-site variations in the bacterial community composition were measured using ANOSIM. The results showed that the inter-site variations in the bacterial community composition were greater than the inner-site variations (R = 1). The PCoA and NMDS analyses provided convincing evidence of variations in the bacterial community composition across the different sampling sites (Fig. 4). The bacterial community structures of sites PTT and AYY were closer to each other. Furthermore, the beta diversity heat map representing an explicit comparison of bacterial communities based on their composition confirmed that the bacterial community composition of site PTT was most closely related to AYY at 0.183 (Fig. 5), whereas the bacterial community composition of site AYY was the most dissimilar to site NYK (0.341), followed by site PAR (0.338). These results were supported by the UPGMA dendrogram (Fig. 6) that showed the relationships for the relative abundance of each sampling site at the phylum level. The created dendrogram consisted of two main clusters. The first cluster formed of sites PTT, AYY, and CCS, with sites PTT and AYY closer to each other but linked together with site CCS. The second cluster was composed of site NYK together with site PAR.

Figure 4 PCoA (A) and NMDS (B) analyses bacterial composition similarity among sampling sites.

CCS, sampling from Chachoengsao province; NYK, sampling from Nakhon Nayok province; PTT, sampling from Pathum Thani province; AYY, sampling from Phra Nakhon Si Ayutthaya province; PAR, sampling from Prachin Buri province.

Figure 5 Beta diversity heat map of the dissimilarity coefficient between each sample.

CCS, sampling from Chachoengsao province; NYK, sampling from Nakhon Nayok province; PTT, sampling from Pathum Thani province; AYY, sampling from Phra Nakhon Si Ayutthaya province; PAR, sampling from Prachin Buri province.

Figure 6 The UPGMA dendrogram of relative abundance at phylum level from each sampling site.

CCS, sampling from Chachoengsao province; NYK, sampling from Nakhon Nayok province; PTT, sampling from Pathum Thani province; AYY, sampling from Phra Nakhon Si Ayutthaya province; PAR, sampling from Prachin Buri province.

Effect of environmental constituents on soil bacterial community and nematode distribution

Regarding the influences of the soil physicochemical parameters on the soil bacterial communities, the analysis presented that the soil pH was the most positively correlated with the members of the Polyangia and most negatively correlated with the members of the Acidobacteriae (Table S2). The Ec and total K content were the most positively correlated with members of the Anaerolineae and the most negatively correlated with members of the Verrucomicrobiae. The OM was the most positively correlated with members of the Polyangia as well as the bacterial community and the most negatively correlated with members of the alpha-proteobacteria. The available N was the most positively correlated with the bacterial community and most negatively correlated with members of the alpha-proteobacteria. Total P and Mn were the most positively correlated with members of the Bacteroidia and the most negatively correlated with members of the Acidobacteriae. Total Ca was the most positively correlated with members of Polyangia and the most negatively correlated with members of the alpha-proteobacteria and Acidobacteriae. Total Mg was the most positively correlated with members of the Anaerolineae and the most negatively correlated with members of the Verrucomicrobiae. Total Fe was the most negatively correlated with members of the Anaerolineae. Total Cu and Zn were the most positively correlated with members of the Anaerolineae and the most negatively correlated with members of the Acidobacteriae.

In addition, after evaluating the effect of the soil physicochemical parameters on phytonematodes and free-living nematodes in the rhizosphere soils, soil pH was the most positively correlated with free-living nematodes and the most negatively correlated with Pratylenchus spp. (Table S2). The Ec was the most positively correlated with Helicotylenchus spp. and the most negatively correlated with Pratylenchus spp. The OM and available N were the most negatively correlated with Tylenchorhynchus spp. The total P was the most positively correlated with Hirschmanniella spp. and the most negatively correlated with Tylenchorhynchus spp. Total K and Mg were the most positively correlated with Helicotylenchus spp. and the most negatively correlated with Pratylenchus spp. In contrast, the total Fe was the most positively correlated with Pratylenchus spp. and the most negatively correlated with Helicotylenchus spp., whereas total Ca was negatively correlated with Pratylenchus spp. Total Mn was the most positively correlated with free-living nematodes and the most negatively correlated with Tylenchorhynchus spp. Total Cu and Zn were the most positively correlated with M. graminicola and the most negatively correlated with Pratylenchus spp. The correlations between soil nematodes and the bacteriome were investigated (Table S3). The results showed that Hirschmanniella spp. were the most positively correlated with members of the Bacteroidia. Pratylenchus spp. were the most positively correlated with members of the Actinobacteria and the most negatively correlated with members of the Polyangia. Tylenchorhynchus spp. were the most negatively correlated with members of the Clostridia and Bacilli. Free-living nematodes in soils were the most negatively correlated with members of the Acidobacteriae.

The influences of the soil physicochemical parameters on phytonematodes and free-living nematodes within roots were evaluated. The results showed that soil pH and Ec were the most positively correlated with M. graminicola and the most negatively correlated with Pratylenchus spp. The OM and total Mn were the most negatively correlated with Tylenchorhynchus spp., whereas total P was the most positively correlated with Helicotylenchus spp. The total contents of K, Ca, and Mg were the most positively correlated with M. graminicola and the most negatively correlated with Pratylenchus spp. Fe concentration was the most positively correlated with Pratylenchus spp. and the most negatively correlated with Tylenchorhynchus spp. Total Cu and Zn were the most positively correlated with Helicotylenchus spp. and the most negatively correlated with Pratylenchus spp. The results of the correlations between nematodes within roots and the bacteriome are provided in Table S3. M. graminicola was the most negatively correlated with the members of the Actinobacteria. Pratylenchus spp. were the most positively correlated with members of the Actinobacteria and the most negatively correlated with members of the Polyangia. In addition, the CCS analysis showed that soil pH, Ec, total Ca, K, Mg, and Fe were factors that affected the bacterial community composition and diversity of phytonematodes and free-living nematodes in the rhizosphere and roots of rice (Fig. S3).

Discussion

Rice is a globally significant agricultural crop cultivated on a massive scale (Gnamkoulamba et al., 2018). Phytonematodes are one of the barriers to improved rice production. Even though over 4,100 species of phytonematodes have been reported, including endo- and ecto-parasites (Decraemer & Hunt, 2006), a mere 29 species have been identified as having a direct correlation with yield reductions in rice production (Bridge, Plowright & Peng, 2005). The occurrence of nematode attacks has the potential to facilitate the infection process of other pathogens (De Waele & Elsen, 2007). Herein, we surveyed and collected rice-associated nematodes from notable paddy fields. Our results demonstrated that the most prevalent genera of phytonematodes associated with rice agriculture in various agroecological zones of Thailand included M. graminicola, Hirschmanniella spp., Pratylenchus spp., Helicotylenchus spp., and Tylenchorhynchus spp. Notably, M. graminicola and Hirschmanniella spp. had their highest density in soil collected from site PTT. In addition, they were both predominant within roots collected from sites CCS and AYY, while the amounts of free-living nematodes in the soil and roots were highest at sites AYY and PAR. Pascual et al. (2014) investigated the widespread presence of nematodes in rice fields in Luzon, the Philippines. They reported that Helicotylenchus, Hirschmanniella, Meloidogyne, Criconema, Xiphinema, Pratylenchus, and Tylenchorhynchus were the more prevalent and abundant genera. The main phytonematodes found in Togo’s rice fields were found in the genera Meloidogyne, Suctellonema, Heterodera, Hirschmanniella, Pratylenchus, and Helicotylenchus (Coyne et al., 2000). Gnamkoulamba et al. (2018) recorded the genera Helicotylenchus, Hirschmanniella, Meloidogyne, and Suctellonema were in both soil and root samples of rice in different agroecosystems in Togo. According to our findings, M. graminicola and Hirschmanniella spp. were reported as the dominant group of nematodes more frequently found within the soils and roots of rice (Bridge, Plowright & Peng, 2005; Eche et al., 2013), with M. graminicola being documented as being highly adapted to flooded environments, leading to better survival in soil environments (Bridge, Plowright & Peng, 2005). Musarrat et al. (2016) found these nematodes in a rice-growing area in Pakistan. The present study detected free-living nematodes in rice roots. This was not surprising, as nematode-fungal pathogen disease complexes have been reported in general with nematode infection. Some species of free-living nematodes are fungivores, which have feeding dispositions on fungi, including fugal plant pathogens, so they can invade plant roots to obtain food (Zhang et al., 2020). Different factors affect the distribution of nematodes, such as the production system, rice variety, intercropping with other crops, and rainfall (Gnamkoulamba et al., 2018). However, the high levels of nematode density and diversity observed in the present study suggest that rice cultivation in Thailand is being confronted with severe endo- and epi-phytotic nematodes. These results could be attributed to the escalating intensification of rice cultivation in Thailand.

The activity of soil-dwelling organisms is influenced by physicochemical soil qualities, whereas the establishment of a nematode population in the soil is influenced by a range of abiotic and biotic factors (Al-Ghamdi, 2021). De Oliveira Cardoso et al. (2012) reported that the physicochemical properties of soil had an impact on the density and structural diversity of nematode communities. Our findings demonstrated that the soil at site PTT had the highest numbers of M. graminicola, Hirschmanniella spp., and Helicotylenchus spp., contained the significantly highest concentrations of total P, Mg, Cu, and Zn. The rice roots at site AYY had the highest number of Hirschmanniella spp. and had the significantly highest levels of soil OM, available N, total Ca, Mn, and pH. Rice roots at site CCS had the significantly highest number of M. graminicola also had the highest soil Ec value and total K content. These results suggested that the establishment of infection by M. graminicola could be significantly associated with soil parameters, including Ec and the levels of total K, P, Mg, Cu, and Zn. Many soil factors (including the quantity of OM, levels of available N, total Ca and Mn, and pH) could support infection by Helicotylenchus spp. These findings appeared to be consistent with other reports that demonstrated the contents of soil elements, such as OM, P, Ca, Mg, and K, supported the establishment of phytonematodes in the soil (Al-Ghamdi, 2021; Dias-Arieira et al., 2021; Leiva et al., 2020). Furthermore, our results showed that a higher soil pH and level of OM supported higher populations of phytonematodes and free-living nematodes. This finding was consistent with Castro et al. (1990) and Al-Ghamdi (2021) who reported that pH and OM played important roles in the proportion of soil nematodes. The distribution of soil nematodes was negatively correlated with the pH; soil acidity affected nematode populations, such as M. incognita and Radopholus similis, that were presented at reduced levels in acidic soils (Davide, 1980; Gade & Hiware, 2017). The soil OM content positively supported the high proportions of free-living nematodes by promoting bacteria and fungi growth, which were essential foods for the nematodes (Cadet & Spaull, 2003).

Understanding the role that the plant rhizosphere soil microbiome plays during PPN infection is considerable and should be investigated. Previous research documented the significance of microbial communities presented in soil in the control of phytonematodes (Silva et al., 2022; Zhou et al., 2019). Thus, the current study conducted an extensive evaluation of bacterial diversity and community composition in soils infested with nematodes, with a focus on the influence of environmental conditions. The findings showed there were no significant differences in bacterial diversity across the sampled locations. Nevertheless, site CCS exhibited the greatest number of detected species and bacterial richness. Our results may indicate an effect of nematode density and diversity on the observed species and bacterial richness. Nematodes have been reported to graze on bacteria, which may affect the bacterial community by accelerating bacterial turnover (Cheng et al., 2016; Djigal et al., 2004). Silva et al. (2022) concluded that bacterial richness in a community was reduced as a response to the numbers of nematodes in infested soil. Furthermore, the present study found that the phyla Acidobacteriota, Proteobacteria, Firmicutes, Actinobacteriota, Myxococcota, Chloroflexi, Verrucomicrobiota, Bacteroidota, Gemmatimonadota, and Desulfobacterota were the top-10 regarding bacterial abundance in all the sampled soils. This data was according to the results from other studies. For example, Vinothini et al. (2024) reported that Proteobacteria, Firmicutes, and Actinobacteria were the dominant bacterial taxa, while Ascomycota, Basidiomycota, and Mucoromycota were prevalent among the fungal taxa in the tomato rhizosphere. Silva et al. (2022) reported that Acidobacteriota, Proteobacteria, Firmicutes, Actinobacteriota, and Gemmatimonadota were the most common bacterial groups in soil samples in Brazil. Masson et al. (2020) found that the Acidobacteriota, Proteobacteria, Actinobacteriota, Verrucomicrobia, Nitrospirae, and Fibrobacteres were the predominant phyla of highly M. graminicola-infested fields in Vietnam. The bacterial phyla Acidobacteriota, Proteobacteria, Actinobacteriota, and Gemmatimonadota were the most abundant in phytonematode-suppressive soils (Harkes et al., 2020). Members of the Acidobacteriota, Proteobacteria, Firmicutes, and Actinobacteriota exhibited broad metabolic diversity and possess the ability to colonize various ecosystems. These bacterial groups possess a multitude of genes associated with stress resistance, carbon degradation, phosphate solubilization, and antibiotic production. These genetic traits contribute to their adaptive capabilities and enable their successful survival in soil, thereby establishing them as dominant microbial groups (Pongsilp & Nimnoi, 2022). In addition, our results demonstrated that the soil pH, OM, Ec, and total Ca, K, Mg, and Fe were factors that might be affecting bacterial diversity. Our result is compatible with prior research that has noted the influence of many soil variables, including pH, OM, N, K, Mg, and Zn concentrations, as well as nutrient availability and hydrocarbon bioavailability, on the variability of bacterial communities and their diverse composition (Achife, Bala & Oyeleke, 2021; Nimnoi & Pongsilp, 2022; Pongsilp & Nimnoi, 2020). By changing the amount of nutrients available in the environment, the relationship between soil minerals and bacteria can influence biogeochemical cycling (Pongsilp & Nimnoi, 2020; Vu et al., 2022). Correlations between the bacterial microbiome and phytonematodes were found, which were greater than in the roots. The numbers of Acidobacteriae, Clostridia, Bacilli, Bacteroidia, Polyangia, and Actinobacteria influenced the proportions of Hirschmanniella spp., Pratylenchus spp., Tylenchorhynchus spp., and free-living nematodes in the soil samples, whereas only the numbers of Polyangia and Actinobacteria affected the numbers of M. graminicola and Pratylenchus spp. within the roots. Castillo, Vivanco & Manter (2017) reported correlations between dominant bacteria and nematode populations. The alpha-proteobacteria, Rhodoplanes, Phenylobacterium, and Kaistobacter have been found to be correlated with the Meloidogyne, while the Bacteroidia and gamma-proteobacteria have been reported to be correlated with the Pratylenchus. Members of the Bacilli, Polyangia, Actinobacteria, and Acidobacteriae are important contributors to ecosystems since they are particularly abundant and ubiquitous in nature, such as in the soil, roots, water, and sediment (Eichorst, Breznak & Schmidt, 2007; Nimnoi, Pongsilp & Lumyong, 2011; Nimnoi & Pongsilp, 2022). Furthermore, they have been noted for their effects on the control of biogeochemical cycles, the degradation of biopolymers, the release of exopolysaccharides, and the encouragement of plant development (Kalam et al., 2020; Matsumoto et al., 2021; Puri, Padda & Chanway, 2018; Puri, Padda & Chanway, 2020). Additionally, they can synthesize diverse natural compounds that produce biomedically and industrially useful chemicals, such as antifungals, antibiotics, and antinematodal agents, which can be applied in regulating and affecting diverse microorganisms in ecosystems (Atta & Ahmad, 2009; Crits-Christoph et al., 2018; Hadjithomas et al., 2015; Mahajan, 2012; Nigris et al., 2018; Parsley et al., 2011; Reichenbach, 2001).

Notably, biocontrol plant diseases and plant growth-promoting bacteria, such as the genera Bryobacter, Acidibacter, Pseudomonas, Burkholderia, Caballeronia, Paraburkholderia, and Sphingomonas, were the predominant bacterial groups identified in the sampling soils in the present investigation. These genera have been found in soils where there has been significant suppression of soilborne disease through biocontrol, including phytonematodes, as well as plant growth promotion. Pseudomonas spp. are chitinolytic and hydrogen cyanide-producing bacteria that can be applied for the biocontrol of nematodes (Ha et al., 2014; Kang, Anderson & Kim, 2018; Lee et al., 2011). Burkholderia, Caballeronia, Paraburkholderia, and Sphingomonas have been stated to function biological nitrogen fixation, increase nutrient uptake, and confer disease resistance against a bacterial pathogen (Matsumoto et al., 2021; Puri, Padda & Chanway, 2018; Puri, Padda & Chanway, 2020). The establishment of Acidibacter in many plant species has been reported in association with soil iron and nutrient cycles, and soil pollution treatments (Huang et al., 2020; Jiao et al., 2018; Liu et al., 2016). The genus Bryobacter has been noted as a beneficial microorganism for leguminous plants by playing roles in the degradation of minerals, promotion of plant growth, nitrogen fixation, and the suppression of plant disease (Li et al., 2023; Luis et al., 2018; Xiao et al., 2017). Nematodes and the microbial community have been reported for their activities in response to environmental impacts and eco system conversions (Briar, Grewal & Somasekhar, 2007; Renčo, Gömöryová & Čerevková, 2020). The present results have provided comprehensive data that could be beneficial for designing an appropriate cultivation method to control rice diseases caused by phytonematodes and to preserve soil quality for sustainable management.

Conclusions

The differences in soil properties, and numbers of nematodes and bacterial communities in the soils sampled reveal the clear impact of biotic and abiotic soil characteristics on ecosystem variables. We demonstrated that M. graminicola, Hirschmanniella, Pratylenchus, Helicotylenchus, and Tylenchorhynchus were the dominant phytonematodes distributed in soil across rice fields in Thailand. The high-throughput sequencing analysis clarified that Acidobacteriota, Proteobacteria, Firmicutes, Actinobacteriota, Myxococcota, Chloroflexi, Verrucomicrobiota, Bacteroidota, Gemmatimonadota, and Desulfobacterota were the predominant bacterial phyla that had established niches in the sampled soils. The numbers of Acidobacteriae, Clostridia, Bacilli, and Bacteroidia influenced the proportions of Hirschmanniella spp., Tylenchorhynchus spp., and free-living nematodes in the soil samples, whereas the numbers of Polyangia and Actinobacteria affected the numbers of Pratylenchus spp. in both the roots and soils. The components of total P, K, Mg, Ca, Cu, and Zn, as well as the pH, Ec, and OM of the soil might be influencing the composition of the bacterial and nematode communities. Our findings provided insights into correlations among rhizosphere microbiota, nematodes, and soil properties, contributing to the potential development of suitable management programs to reduce phytonematodes in rice production systems.

Supplemental Information

Supplemental Information 1 Morphometrics of adult females of plant-parasitic nematodes obtained from roots and rhizosphere soils of Oryza sativa L. cv. Khao Dawk Mali 105

DGO = dorsal pharyngeal gland opening; * Mean ± SD (n = 25) ** second-stage juveniles of Meloidogyne graminicola

Supplemental Information 2 Spearman’s (r) correlations of abiotic and biotic factors

*Data shown in format of r (P-value) **Correlation is significant at the 0.05 level

Supplemental Information 3 Spearman’s (r) correlations of top 10 bacterial class and nematodes in soils and roots

*Data shown in format of r (P-value) **Correlation is significant at the 0.05 level *** Correlation is significant at the 0.01 level

Supplemental Information 4 Identification of nematodes obtained from roots and rhizosphere soils of Oryza sativa L. cv. Khao Dawk Mali 105 using a set of species-specific primers (Mg-F3/ Mg-R2) for M. graminicola

Lane 1: marker (100 bp DNA Ladder RTU); M, Lane 2: Chachoengsao; CCS, Lane 3: Nakhon Nayok; NYK , Lane 4 : Prachin Buri; PAR, Lane 5: Pathum Thani; PTT, Lane 6: Phra Nakhon Si Ayutthaya; AYY, Lane 7: M. incognita; Mi, Lane 8: negative control; Ck.

Supplemental Information 5 LEfSe analysis at multi taxonomic levels comparing bacterial community composition of each sampling site

(A) Histogram of the LDA scores generated for groups with differential abundance among the bacterial communities of each site. (B) Between-group analysis, a double asterisk represents vary significant variation (p < 0.01), and a single asterisk represents significant variation (p < 0.05).

Supplemental Information 6 CCA of bacterial data and soil physicochemical characteristics

The influence of environmental factors on bacterial community structure is indicated by green line. The length of each line represents the impact of the corresponding environmental factor on the distribution of bacterial community. The longer line indicated the greater effect. OS, Observed species; Acido, Acidobacteriae; Gamma, Gamma-proteobacteria; Alpha, Alpha-proteobacteria;C; Ver, Verrucomicrobiae; Bacter, Bacteroidia; Anaer, Anaerolineae; Actino, Actinobacteria; MeloS, M. graminicola in rhizosphere soils; HirS, Hirschmanniella spp. in rhizosphere soils; PraS, Pratylenchus spp. in rhizosphere soils; HelS, Helicotylenchus spp. in rhizosphere soils; TylS, Tylenchorhynchus spp. in rhizosphere soils; FlS, Free living nematode in rhizosphere soils; MeloR, M. graminicola within roots; HirR, Hirschmanniella spp. within roots; PraR, Pratylenchus spp. within roots; HelR, Helicotylenchus spp. within roots; TylR, Tylenchorhynchus spp. within roots; FlR, Free living nematode within roots; CCS, sampling from Chachoengsao province; NYK, sampling from Nakhon Nayok province; PTT, sampling from Pathum Thani province; AYY, sampling from Phra Nakhon Si Ayutthaya province; PAR, sampling from Prachin Buri province.

Supplemental Information 7 Raw data of OTUs

CCS, sampling from Chachoengsao province; NYK, sampling from Nakhon Nayok province; PTT, sampling from Pathum Thani province; AYY, sampling from Phra Nakhon Si Ayutthaya province; PAR, sampling from Prachin Buri province.

Supplemental Information 8 Raw data bacterial composition

CCS, sampling from Chachoengsao province; NYK, sampling from Nakhon Nayok province; PTT, sampling from Pathum Thani province; AYY, sampling from Phra Nakhon Si Ayutthaya province; PAR, sampling from Prachin Buri province.

Supplemental Information 9 Raw data bacterial richness and diversity

CCS, sampling from Chachoengsao province; NYK, sampling from Nakhon Nayok province; PTT, sampling from Pathum Thani province; AYY, sampling from Phra Nakhon Si Ayutthaya province; PAR, sampling from Prachin Buri province.

Supplemental Information 10 Raw data distribution of bacterial genus

CCS, sampling from Chachoengsao province; NYK, sampling from Nakhon Nayok province; PTT, sampling from Pathum Thani province; AYY, sampling from Phra Nakhon Si Ayutthaya province; PAR, sampling from Prachin Buri province.

Supplemental Information 11 Raw data phylum relative abundance

CCS, sampling from Chachoengsao province; NYK, sampling from Nakhon Nayok province; PTT, sampling from Pathum Thani province; AYY, sampling from Phra Nakhon Si Ayutthaya province; PAR, sampling from Prachin Buri province.

Supplemental Information 12 Raw data of Tags and Clean data of Tags

CCS, sampling from Chachoengsao province; NYK, sampling from Nakhon Nayok province; PTT, sampling from Pathum Thani province; AYY, sampling from Phra Nakhon Si Ayutthaya province; PAR, sampling from Prachin Buri province.

Supplemental Information 13 Raw data of top ten bacterial class

CCS, sampling from Chachoengsao province; NYK, sampling from Nakhon Nayok province; PTT, sampling from Pathum Thani province; AYY, sampling from Phra Nakhon Si Ayutthaya province; PAR, sampling from Prachin Buri province.

Supplemental Information 14 Number of nematode in soils and roots of each site

CCS, sampling from Chachoengsao province; NYK, sampling from Nakhon Nayok province; PTT, sampling from Pathum Thani province; AYY, sampling from Phra Nakhon Si Ayutthaya province; PAR, sampling from Prachin Buri province.

Supplemental Information 15 Raw data of soil physicochemicals

CCS, sampling from Chachoengsao province; NYK, sampling from Nakhon Nayok province; PTT, sampling from Pathum Thani province; AYY, sampling from Phra Nakhon Si Ayutthaya province; PAR, sampling from Prachin Buri province.

Additional Information and Declarations

Competing Interests

Author Contributions

Data Availability

The authors declare there are no competing interests.

Pongrawee Nimnoi conceived and designed the experiments, performed the experiments, analyzed the data, prepared figures and/or tables, authored or reviewed drafts of the article, and approved the final draft.

Patawee Pirankham performed the experiments, analyzed the data, prepared figures and/or tables, authored or reviewed drafts of the article, and approved the final draft.

Kittipong Srimuang performed the experiments, authored or reviewed drafts of the article, and approved the final draft.

Pornthip Ruanpanun conceived and designed the experiments, performed the experiments, analyzed the data, prepared figures and/or tables, authored or reviewed drafts of the article, and approved the final draft.

The following information was supplied regarding data availability:

The sequence data is available at the Sequence Read Archive of the NCBI: PRJNA977457.

The raw data are available in the Supplemental Files.

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
