# Peer review of "Insights into soil nematode diversity and bacterial community of Thai jasmine rice rhizosphere from different paddy fields in Thailand"

_PeerJ, doi:10.7717/peerj.17289_

## Round 0.1 · original submission · Minor Revisions

Thank you for submitting your above-mentioned paper. The evaluation procedure of your contribution is now finished. In view of the comments of the referees, some minor revision is required. Therefore, we suggest you address each comment and revise the manuscript accordingly.

**Language Note:** The review process has identified that the English language must be improved. PeerJ can provide language editing services - please contact us at copyediting@peerj.com for pricing (be sure to provide your manuscript number and title). Alternatively, you should make your own arrangements to improve the language quality and provide details in your response letter. – PeerJ Staff

·

Basic reporting

The manuscript “Insights into soil nematode diversity and microbiome of Thai Jasmine rice rhizosphere from different paddy fields in Thailand” studied rice nematodes, soil parameters, and bacterial communities across five rice-growing regions to find correlations is a well-presented manuscript.

The language of the paper is good, except for minor grammar issues. I request that the authors get the manuscript edited/checked for grammar and language to improve the readability of the manuscript as several minor errors are present throughout the manuscript.

For e.g.- ‘significant’ to be changed to ‘significantly’ in line 32 of the abstract, ‘phytonematodes’ to ‘phytonematode’ in line 83.

‘Goals were aimed at’ (lines 82-83) are redundant and must be stated as ‘Goals were to investigate’
Line 200- Add “At” before all sampled locations

There are several mistakes like this and it's not possible to point all of them here, hence authors should please get it checked.

Experimental design

The research question of investigating the association between the phytonematode population density and the rice root-associated microbiome from the same rice cultivar under different natural environmental conditions is well framed.

The experimental design is good, and the experimental and statistical methods employed to investigate the research problem are adequate.

Validity of the findings

The authors found that M. graminicola, Hirschmanniella spp were major nematodes, and that soil parameters (total 30 P, Cu, Mg, and Zn) were important in affecting the abundance of both nematodes. Classes of the Acidobacteriae, Clostridia, Bacilli, and Bacteroidia influenced the proportions of Hirschmanniella spp., Tylenchorhynchus spp., and free-living nematodes in the sampling dirt, whereas the numbers of classes of the Polyangia and Actinobacteria affected the amounts of Pratylenchus spp. in both roots and soils.

The findings are valid. All underlying data have been provided, are robust, and statistically sound.

However, the conclusions are strong and need to be toned down a bit. All that the evidence points to is correlations between various nematodes, bacterial communities, and soil composition. To conclude (Lines 510-512) “The components of total P, K, Mg, Ca, Cu, and Zn, as well as the pH, Ec, and OM of the soil influenced the composition of the bacterial and nematode communities.” is a strong conclusion, since no such experiment was carried out to functionally prove this correlation.

Authors may suggest that it appears that the soil components such as P. K. Mg, Ca, Cu, Zn, and so on might be influencing the composition of the bacterial and nematode communities” at various places in the manuscript.

·

Basic reporting

The manuscript entitled on Insights into soil nematode diversity and microbiome of Thai Jasmine rice rhizosphere from different paddy fields in Thailand, has been well focused to understand the diversity of nematode and bacterial community. However, the title highlights on diversity of microbiome. But all the work has been concentrated only on bacterial diversity. Hence, the title has to be changed accordingly. There are grammatic mistakes in the manuscript. It has been highlighted . Those changes has to be amended to increase the clarity. Overall, the manuscript has been well written and the objectives has been framed well. Some of the recent publication published related to community composition of bacterial diversity published in 3 Biotech in correlation with nematode in tomato rhizosphere can be included.

Experimental design

Experimental design has been well planned. However the methodology pertaining to rhizosphere soil sampling has not been included in the methodology. Metagenomics study has been well planned. The work has novelty and also gains significance. the manuscript meet the standards. Primer details on the nematode has to be included.

Validity of the findings

Discussion part is too lengthy it can be edited. The results of the nematode identification through microscope and molecular confirmation has to be discussed. Photos of the identified nematodes has to be included in the manuscript. Discussion part has to be highly focused based on the diversity of bacterial community and nematode proliferation in the soil and inside the root. This part is very fluid and need more attention. Conclusion has been well defined.

Additional comments

The corrections mentioned in the PDF version of the manuscript has to be taken care. Discussion part has to be well focused. Why there is increase in particular genera in certain location of Thai. It has to be also linked with the population diversity of bacterial population. The broad name of microbiome has to be changed as diversity of bacteriome.

---

## Round 0.2 · accepted · Accept

I am pleased to inform you that reviewers have given positive feedback on your manuscript. Congratulations, and thank you for your submission.

·

Basic reporting

The authors have satisfactorily addressed my comments, including revising the manuscript for English Language. I recommend that the manuscript may now be accepted for publication.

Experimental design

NA

Validity of the findings

NA

·

Basic reporting

Language has been improved to the maximum extent.

Experimental design

Perfectly All right

Validity of the findings

This is an emerging science in nematology. Change in bacteriome in the rice rhizosphere and correlation of it with the plant parasitic nematodes has added insights to the young researchers.

Additional comments

Author has taken care to address all the comments in a meticulous way.